# Exploring the Relation between Atopic Diseases and Lifestyle Patterns among Adolescents Living in Greece: Evidence from the Greek Global Asthma Network (GAN) Cross-Sectional Study

**DOI:** 10.3390/children8100932

**Published:** 2021-10-18

**Authors:** George Antonogeorgos, Kostas N. Priftis, Demosthenes B. Panagiotakos, Philippa Ellwood, Luis García-Marcos, Evangelia Liakou, Alexandra Koutsokera, Pavlos Drakontaeidis, Dafni Moriki, Marina Thanasia, Maria Mandrapylia, Konstantinos Douros

**Affiliations:** 1Allergology and Pulmonology Unit, 3rd Paediatric Department, National and Kapodistrian University of Athens, 12462 Athens, Greece; kpriftis@otenet.gr (K.N.P.); drliakouevangelia@yahoo.gr (E.L.); alexandra.koutsokera@gmail.com (A.K.); pauldrakos@hotmail.com (P.D.); dafnimoriki@yahoo.gr (D.M.); marinarod9422@gmail.com (M.T.); marmand24@outlook.com (M.M.); costasdouros@gmail.com (K.D.); 2Department of Nutrition and Dietetics, School of Health Sciences and Education, Harokopio University, 17676 Athens, Greece; dbpanag@hua.gr; 3Department of Pediatrics, Child and Youth Health, Faculty of Medical and Health Sciences, University of Auckland, Auckland 1023, New Zealand; p.ellwood@auckland.ac.nz; 4Pediatric Allergy and Pulmonology Units, “Virgen de la Arrixaca” University Children’s Hospital, Biomedical Research Institute of Murcia, University of Murcia, Network of Asthma and Adverse and Allergic Reactions (ARADyAL), IMIB-Arrixaca, 30394 Murcia, Spain; lgmarcos@um.es

**Keywords:** fruits, vegetables, pulses, atopy, adolescents, lifestyle, pattern

## Abstract

Introduction: Diet and physical activity might be associated with the risk of allergic diseases in childhood. However, evidence in literature is sparse and diverse. We aim to examine the associations between four healthy dietary consumption pattern drinks, plus the adherence to a physically active lifestyle with atopic diseases (asthma, allergic rhinitis and eczema) in adolescence and their relative importance. Methods: A total of 1934 adolescents (921 boys, 47.5%) and their parents completed a validated questionnaire assessing atopic diseases’ symptoms prevalence in the past 12 months, as well as nutritional and physical activity information. Four healthy dietary and one physical active lifestyle patterns were identified and logistic regression was applied to assess their relation with allergic diseases. Results: A high weekly consumption of fruits, vegetables and pulses and low consumption of unhealthy foods was negatively associated with all atopic symptoms while adherence to a physical active lifestyle was inversely associated with asthma and allergic rhinitis symptoms and dairy products with asthma and eczema symptoms in the past 12 months after adjustment for several confounders (all *p* < 0.05). Fruits, vegetables and pulses consumption per week emerged as the most important lifestyle pattern negatively associated for all atopic diseases, after the adjustment for all the remaining lifestyle patterns and confounders (all *p* < 0.05) Conclusions: Our findings suggest that a high fruit, vegetable and pulse intake should be the first lifestyle intervention every clinician and public health care worker evolving in the management of atopic adolescents should encourage and promote.

## 1. Introduction

Atopic diseases in Europe are among the major causes of morbidity in children due to their high prevalence; however, evidence about the implicated factors in their pathogenesis is sparce and diverse. Results from the ISAAC (The International Study on Asthma and Allergy) study, the first worldwide study to measure the occurrence of asthma, allergic rhinitis and eczema in childhood, showed different geographical and temporal patterns among the participating European countries [1]. Greece is one of the countries with the lowest prevalence of allergies among the European countries. However, a similar rising trend, observed in westernized societies—a dramatic rise in the 1980s and 1990s, followed by a plateau in the 2000s and a slight decline afterwards—has been reported for asthma, while for allergic rhinitis and eczema the reported evidence is conflicting for both Greece and other European countries [2,3,4].

This variation in the prevalence of atopic diseases in childhood could be attributed to their complex association with several lifestyle factors, such as physical activity and nutrition. Regarding nutrition, food items, food groups and dietary patterns have been inversely related with atopic diseases. There is evidence suggesting that children who are high consumers of food groups such as vegetables and fruits or have a high adherence to healthy dietary patterns with less consumption of fast-food and a high consumption of fruits and vegetables are less likely to have asthma and rash symptoms as well as a lower allergic sensitization [5,6,7]. Similar findings have been reported for allergic rhinitis also [8,9]. Moreover, recent studies have documented that children with reduced physical activity could suffer more from asthma and all the other atopic diseases [10,11]. These associations could be attributed to the antioxidative and the anti-inflammatory effect of a healthy diet and increased physical activity. Oxidative stress and inflammation play a central role in pathogenesis and the clinical presentation for atopic diseases. A diet rich in fruits, vegetables as well as other healthy foods such as dairy products could provide a variety of antioxidants and anti-inflammatory agents sufficient enough to interfere with the atopic pathophysiological mechanisms; thus, reducing the severity and the duration of the allergic symptomatology [12,13,14]. However, the magnitude of the effect of this evidence remains to be defined.

Enlightening the relationship between lifestyle factors and atopic diseases in childhood could provide public health care workers with low-cost and effective intervention means, which could help reduce the burden of these chronic diseases of childhood and could also improve the quality of life of children and their families. Moreover, the variations in the prevalence of childhood atopic diseases in recent decades strengthen the need for a continuous and extensive study of the influence of lifestyle characteristics on them. In this context, the Global Asthma Network study (GAN) was established and designed to assess, worldwide, the symptom prevalence of asthma, eczema and rhinitis and to identify several environmental factors associated with them in children aged 6/7 and 13/14 years old [15]. Using data from the Greek part of the GAN study, we aim to examine (a) the associations between four healthy dietary consumption patterns—namely, a high consumption of fruits, vegetables and pulses, high consumption of starchy products, high consumption of dairy products and low consumption of fast-food, sweets and soft drinks—, plus the adherence to a physically active lifestyle, asthma, eczema and allergic rhinitis symptoms in the past 12 months; (b) to evaluate the relative importance of each of these lifestyle factors compared to the others.

## 2. Materials and Methods

### 2.1. Design

This was a cross-sectional study, part of GAN Phase I, which was an international project aimed to monitor the worldwide prevalence, severity, management and risk factors of asthma and other atopic diseases such as allergic rhinitis and eczema, in two time periods of childhood: 6/7 and 13/14 years old [16]. The Greek part of the study included only adolescents aged 13/14 years.

### 2.2. Setting and Sample

The study took place in the greater metropolitan area of Athens, Greece, from February to March 2020, in a high-school setting; twenty high-schools were selected by convenience sampling, from a list that was provided by the Secondary Education Office in Athens. All children in the 1st and 2nd grades of each high-school were asked to participate. Schools for children with special educational needs or disabilities were excluded. In total, 2560 child–parent/guardian pairs were asked to participate. Of them, 1934 adolescents (921 boys, 47.5%, and 1013 girls, 52.5%), with mean age and Standard Deviation (SD) of 12.7 (0.6) years and their parents/guardians (25.4% fathers with mean age 49.1 (5.5) years and 74.6% mothers with mean age 45.4 (4.8) years) agreed to participate (participation rate 76%).

### 2.3. Bioethics

The study was approved by the ethics committee of the National and Kapodistrian University of Athens (decision number: 214/13-12-19). For the accomplishment of the study, permission was issued by the Ministry of Educational Affairs (decision number: 10053/24-01-2020).

### 2.4. Measurements

The GAN study included two standardized questionnaires: one that was completed by the adolescents during schooltime and one intended for parents/guardians to complete at home (adult questionnaire). The adolescents’ questionnaire included several questions about symptoms of asthma, eczema and allergic rhinitis as well as other questions regarding their dietary intake, their physical activity and information about their family and home environment [17,18,19]. Specifically, current asthma was defined as a positive answer to the question “Have you had wheezing or whistling in the chest in the past 12 months?” Similarly, current rhinitis was defined as a positive answer to the question “In the past 12 months, have you had a problem with sneezing or a runny or blocked nose when you DID NOT have a cold or the flu?” and current eczema as a positive answer to the question “Have you had this itchy rash at any time in the past 12 months?”. Moreover, adolescents were asked if they had any siblings and, if so, how many. Adolescents’ parents or guardians were also asked to report if they had a history of atopic diseases (asthma, eczema or allergic rhinitis), if there was visible moisture or mold spots on the walls or ceiling of their homes and whether they smoked. The participating parental/guardian educational level was recorded in three categories (primary (compulsory education/9 years), secondary (non-compulsory/3 years) or tertiary (university/college/post-graduate studies)). Due to the small number of parents who had primary education (*n* = 24), we merged this level with the next one (secondary education), creating a dichotomous variable (i.e., parental primary/secondary educational level vs. tertiary).

The GAN questionnaire included a validated 22-item Food Frequency Questionnaire (FFQ) assessing the past 12-month consumption frequency of 22 food groups or food items [20]. More specifically, adolescents answered 22 questions related to the consumption frequency of 10 food groups, namely, meat, sea-food—including fish—, fruits, cooked vegetables (green and root), raw vegetables (green and raw), pulses (peas, beans, lentils), cereals, dairy (cheese and yoghurt), sugar (including lollies/candies/sweets), fast-food (excluding burgers), and fizzy or soft drinks and 12 food items (bread, pasta, rice, margarine, butter, olive oil, milk, eggs, nuts, potatoes, fast-food (burgers)), choosing one of the three following options: never or only occasionally, once or twice per week and most of all days for the past 12 months. The consumption of meat, seafood, fruits, cooked and raw vegetables, pulses, cereals, bread, pasta and rice, olive oil, milk and dairy products, nuts, fast-food, sweets and candies and soft drinks was further recoded into two categories, in order to classify adolescents into high (most or all of the days in the past 12 months) vs. low (never up to twice per week in the past 12 months) consumption per week. Regarding physical activity of the participating adolescents, this was assessed through three detailed questions—(a) the number of occasions per week engaging in vigorous physical activity up to the point of breathing hard with the following possible answers: never or occasionally, once or twice per week and three or more times per week; (b) the number of hours a day watching television/film or videos and (c) the number of hours per day engaging in computer and internet activities (gaming, chatting, Facebook, YouTube, etc.) with the following possible answers: less than one hour, over one and up to three hours, over three and up to five hours and more than five hours per day. Likewise, these questions were further recoded and adolescents were classified into two categories regarding their level of vigorous physical activity (high vs. low if they were engaging in vigorous physical activity until it was difficult to breathe for at least three times per week vs. up to twice per week) and daily sedentary activity due to TV watching or computer and internet activities (increased vs. decreased if adolescents were engaging for more than three hours per day or more vs. up to three hours daily).

Furthermore, adolescents, according to their answers in the physical activity and dietary assessment questions, were categorized into five lifestyle patterns. Adolescents who were classified as having a high physical activity level per week and low TV, computer and internet daily engagement were characterized as adherent to an active physical activity lifestyle and as high consumers of the following food groups: (a) fruits, vegetables and pulses; (b) carbohydrates (cereal, bread, pasta, rice), and (c) all dairy products (milk, cheese, yogurt) if they were consuming this food most of all days per week. Moreover, they were also categorized as low consumers of unhealthy food (fast-food, sweets, lollies and soft drinks) if they were consuming up to twice per week any of the aforementioned food groups.

The height and the weight of the participating children were measured and children’s body mass index (BMI) was calculated in order to classify them as normal weight, overweight and obese, using the International Obesity Task Force (IOTF) classification [17]. The three categories were further categorized into overweight and/or obese vs. normal-body-structure adolescents. These cut-off points were based on health-related adult definitions of overweight (25 kg/m^2^) and obesity (30 kg/m^2)^ and adjusted to the age and sex of children. In particular, we measured standing height to the nearest 0.1 cm with a Raven Minimeter (Raven Equipment Limited, Essex, UK) after students had removed their shoes and body weight to the nearest 0.1 kg on calibrated digital scales (Seca, Hanover, MD, USA).

### 2.5. Statistical Analysis

Continuous variables are presented as mean and standard deviation (SD), and categorical variables are presented as absolute and relative frequencies. Pearson Chi-square and Student’s *t*-test were applied in order to examine for univariate differences among the several qualitative and quantitative physical activity and dietary characteristics of adolescents and current asthma, allergic rhinitis and eczema symptoms status, respectively. Moreover, simple and adjusted logistic regression models were applied to estimate the adolescents’ odds and the corresponding 95% Confidence Intervals (95% CI) of allergic outcomes based on the following lifestyle factors: (a) adherence to an active physical activity lifestyle; (b) high consumption of fruits, vegetables and pulses per week (c); high consumption of carbohydrates (bread, pasta and rice) per week, (d) high consumption of dairy (milk, yogurt and cheese) per week and (e) low consumption of unhealthy foods (fast-food, sweets and soft drinks) per week. Several well-known confounders based on the related literature were included in the adjusted models, i.e., sex, BMI, parental atopic history, parental smoking, pet ownership, having an older sibling, cooking with fuels, and indoor exposure to dampness and/or mold. Furthermore, the association of every lifestyle factor, adjusted for the effect of all the other ones and the aforementioned confounders with the allergic outcomes, was also examined by applying multivariable logistic regression analyses. Deviance residuals were calculated in order to evaluate all logistic models’ goodness-of-fit. All reported probability values (*p*-values) were based on two-sided tests and compared to a significant level of 5%. STATA 14 software was used for all the statistical calculations (STATA Corp., College Station, TX, USA).

## 3. Results

From the total of the 1934 adolescents who constituted the study sample, 133 reported to have had at least an episode of wheezing or whistling in the chest in the past 12 months, 491 that had a problem with sneezing or runny nose without cold symptoms in the past 12 months and 173 that they had an itchy rash in the past 12 months. These numbers corresponded to a prevalence of symptoms of current asthma of 6.9%, in symptoms of current rhinitis of 25.3% and a prevalence of current eczema of 8.9%. The prevalence of overweight or obese adolescents was estimated to be 32.3% (625 adolescents). More information about the general characteristics of the study sample can be found elsewhere [16]. The univariate association between sex, age and several anthropometric, parental, family, physical activity and dietary characteristics and the allergy outcomes is presented in Table 1. Significantly more adolescents with parents that had a history of any atopic disease (asthma, allergic rhinitis and eczema) had symptoms of asthma and allergic rhinitis in the past 12 months (14.2% vs. 7.6% and 65% vs. 55.3%, *p* = 0.01 and *p* = 0.038, respectively), while having at least one older sibling was associated with a lesser presence of rash symptoms at the same time (49.1% vs. 57.9%, *p* = 0.026). Regarding physical activity patterns, watching television more than three hours per day every week and engaging in several computer activities (computer or video gaming) for more than three hours per day every week, were related to higher percentages of adolescents with current asthma and current rhinitis (29.8% vs. 25.0% and 58.3% vs. 54.7%, *p* = 0.04 and *p* = 0.007, respectively). Moreover, a high dietary intake of fruits, vegetables and pulses (most or all day per week consumption) was significantly associated with lesser current asthma symptoms (all *p* < 0.05) and a high dietary intake of cooked vegetables and pulses, while a low dietary consumption of sweets, candies and lollies (up to twice per week) was associated with lesser current allergic rhinitis symptoms (all *p* < 0.05). Conversely, a high dietary consumption of cereal and bread was related to higher current eczema symptoms, whereas a high consumption of pasta was related to a higher amount of asthma symptoms (all *p* < 0.05).

The univariate association of the five lifestyle patterns of the adolescents with allergic outcomes is presented in Table 2. Adolescents who were adherent to an active physical activity lifestyle pattern and were high consumers of dairy products per week had a significantly lesser prevalence of current asthma and rhinitis, where a high dietary intake of fruits, vegetables and pulses per week was significantly negatively related to all allergic outcomes of the study (current asthma, rhinitis and eczema). Finally, a lower weekly consumption of fast-food (consumption up to twice per week) was also significantly associated with lesser percentages of all three allergic outcomes, also (all *p* < 0.05).

In order to further explore the associations between the lifestyle factors with the allergic outcomes of the study, univariate and multivariate logistic regression analyses were applied. Two types of models were created for all the study outcomes (current asthma, current allergic rhinitis and current eczema): (a) the crude ones with the lifestyle factors as the only independent variable and (b) the adjusted ones, which included, apart from the assessed lifestyle factors, the following confounders: sex, obesity status, parental atopic history, pet ownership, parental smoking, having an older sibling and cooking with fuels. Results are presented in Table 3. Adolescents who were having an active physical activity lifestyle were 40% less likely to exhibit any asthma or rhinitis symptoms in the past 12 months, while a high consumption of dairy products was significant associated with a 40% and 30% less likelihood of having any asthma or rash symptoms in the same period. Most important, a high dietary intake of fruits and a low dietary intake of fast-food per week was significantly inversely related to the appearance of any allergic symptoms in the past 12 months (all *p*’s < 0.05).

Finally, the effect of each lifestyle factor on current asthma, rhinitis and eczema symptoms was adjusted for the presence of all the remaining lifestyle factors and the aforementioned confounders, in five multivariate logistic regression models for each allergic outcome. The results of the analyses are shown in Table 4. Asthma symptoms in the past 12 months were significantly negatively associated with a high consumption of fruits, vegetables and pulses and a high consumption of dairy products per week (OR: 0.27, 95% CI: (0.12–0.60) and 0.61 (0.42–0.89), respectively). Regarding allergic rhinitis symptoms, similar significant inverse associations were observed for the association of a high consumption of fruits, vegetables, pulses and dairy products and with a low consumption of unhealthy foods (OR: 0.65, 95% CI: (0.53–0.80)), while current eczema was negatively related with the adherence of an active physical activity lifestyle and a high consumption of fruits, vegetables and pulses (OR: 0.59, 95% CI: (0.38–0.91) and 0.46 (0.25–0.83), accordingly).

## 4. Discussion

In this cross-sectional analysis of a representative sample of Greek adolescents aged 13–14 years old, we found significant negative associations between a physically active lifestyle and current asthma and rhinitis symptoms, between the high consumption of dairy products and current asthma and eczema symptoms, while the high consumption of fruits, vegetables and pulses and the low consumption of unhealthy foods was inversely related with all allergic outcomes, after adjustment for several confounders. When the association of all the lifestyle factors with the atopic diseases was tested, the association between the high consumption of fruits, vegetables and pulses remained negatively associated with all atopic diseases, while dairy products were negatively associated with the presence of asthma and rhinitis symptoms in the past 12 months and the low consumption of unhealth foods had the same inverse association with current rhinitis. Finally, adherence to a physically active lifestyle pattern was adversely associated with eczema symptoms in the past 12 months. To the best of our knowledge, this was the first study that examined the association of several dietary and activity patterns at the same time, enabling the assessment of their relative significance. Thus, we provided novel evidence that could allow for the better design of preventive lifestyle intervention strategies for each allergic outcome and, on a clinical level, a practical approach for clinicians to advise their atopic patients.

One of the most interesting findings of our study was the negative relative significance of the high consumption of fruits, vegetables and pulses in the past 12 months with all the atopic outcomes. The high consumption of these food groups retained for its significance not only when several confounders were entered in the model, but also when this association was adjusted for all the rest of lifestyle factors, regarding both diet and physical activity. The main pathophysiological mechanism that fruits, vegetables and pulses beneficially modified were the occurrence and the severity of atopy through their interfering with systematic inflammatory responses due to their high concentration of various anti-oxidative and anti-inflammatory components. Fresh fruits, vegetables and pulses are among the richest dietary sources of a myriad of antioxidants such as vitamins C, E and β-carotene, flavonoids, isoflavonoids and polyphenolic compounds. There are many studies reporting that oxidative stress is elevated in asthmatic children and adults and increases even more during asthma exacerbations [21,22]. Vitamin C and E are major antioxidants that line the surface of the airways and protect the respiratory epithelium cells from oxidative stress. Low vitamin C levels have been associated with severe asthma symptoms and the obstruction of airways in children with persistent asthma or wheezing [23,24]. Moreover, vitamin E has been related with spirometric indices of lung function in children with moderate asthma [25]. Similar findings have been reported for adults [26,27]. Carotenoids and, especially, lycopene and flavonoids, are some more antioxidant compounds found in high concentrations in fruits and vegetables. The oral intake of lycopene has been found to reduce oxidative stress in bronchi, and to lessen airways’ smooth muscle contraction and mucus hypersecretion [28,29]. The observed negative association can be attributed to the beneficial combination of the intake of all of these important elements. Many studies support our results [30,31,32].

Furthermore, our study expanded the negative association of fruit, vegetable and pulse consumption to the occurrence of eczema and allergic disease symptoms. Inflammation and oxidative stress are also an important pathophysiological mechanism for these atopic diseases, yet evidence about the reported associations is sparse. The majority of the published research supported the beneficial association between a high antioxidant consumption and lesser allergic and eczema symptoms. In a case–control study of school-aged children living in Norway, children with asthma and allergic rhinitis had lower serum levels of blood antioxidants compared to healthy subjects [13]. Using data from the ISAAC study, Seo et al. reported that allergic rhinitis symptoms were inversely associated with the intake of antioxidant-related nutrients, and, especially, with the intake of vitamin C [14]. Regarding eczema, a discrepancy was observed about the role of the consumption of these food groups. Although there are birth-cohort studies that report that the adherence to a dietary pattern high in the consumption of fruits, vegetables and pulses during pregnancy is linked to reduced odds of wheezing and eczema in the offspring in 1, 1.5 and 4 years of life, some others child–mother cohort studies do not support this finding [33,34]. However, other cross-sectional studies have reported analogous findings to our study [35,36]. Finally, pulses is another important food group rich in niacin, zinc, folic acid as well as isoflavonoids [37]. As it was demonstrated in a study of 50 Korean children suffering from eczema, the levels of folic acid, zinc and niacin were lower compared to the recommended normal blood values [38]. Similarly, zinc deficiency was also observed in children with persistent asthma [23]. Thus, the healthy dietary pattern of the high consumption of fruits, vegetables and pulses was not only negatively associated with all the allergic outcomes, but its association remained significant when it was adjusted for all the other dietary and physical activity patterns. This result intensified the importance of fruit, vegetable and pulse consumption compared with other dietary and lifestyle interventions as a dietary pattern with multiple benefits for an adolescent with atopic diseases.

The second important finding of our study was the negative association of a high dairy consumption with current asthma and allergic rhinitis symptoms. Since there is a common belief that milk consumption enhances mucus and exacerbates asthma symptoms, it is of high importance to present evidence about the beneficial role of milk and dairy consumption in asthma and allergic diseases [39]. Although there is abundant evidence about the protective effect of raw, unpasteurized cow’s milk consumption on asthma and allergies, there is not sufficient evidence about the relation of common, pasteurized milk or dairy products consumption with allergic diseases. The GABRIELA study demonstrated with the use of a standardized questionnaire and blood testing that pasteurized and raw milk consumption lowered as much as half the likelihood of developing asthma in a sample of 7606 school-aged children living in rural regions of Germany, Austria and Switzerland [40]. This protective effect was even more evident in the results of the prospective cohort study PRISMA, in which the frequent intake of products containing milk such as full cream, milk and butter were linked to a lower risk of developing asthma in pre-school children [41]. A variety of mechanisms are implicated in the inverse association of dairy products with asthma, allergic rhinitis, and eczema. Dairy products are rich in fat-soluble vitamins D and E, which play an important role in the regulation of inflammatory processes and, particularly, in the lung parenchyma and the airways [42]. It is known that vitamin D deficiency the alters airway structure and lung function in mice and it has been associated with atopic inflammatory severity markers such as increased levels of Immunoglobulin E (IgE) and eosinophilia, as well as increased hospitalization rates in children living in Costa Rica [43,44]. Moreover, dairy fat is a source of short-chain saturated fatty acids (SCSAs), which influence the regulation of various inflammatory pathways due to their anti-inflammatory effect [45]. Finally, a high dairy fat consumption has been associated with a higher dietary intake of ω-3 fatty acids, which are also a component of dairy fatty acids with an anti-inflammatory capacity.

Our findings regarding the inverse association between adherence to an active physical activity lifestyle and the low consumption of unhealthy foods (fast-food, sweets, soft drinks) with current asthma, allergic rhinitis symptoms and atopic dermatitis, after adjustment for several confounders, were in line with many studies in the relevant literature. Fast-food, soft drinks and sweets are foods with a high caloric intake derived from sugars and fats and a poor nutritional value. Moreover, their fats mainly consist of trans- and saturated fatty acids and—in combination with simple sugars—can promote the arachidonic acid‘s metabolism that increases the levels of pro-inflammatory cytokines [46]. Another mechanism that the intake of fast-food could interfere with by the pathophysiological mechanism of atopy is through the alteration of the gut microbiota balance [47]. Gut microbiota play an essential role in the inflammatory environment, which reflects to the general immunity status and, subsequently, to the airways [48]. Regarding the relation of physical activity with atopy, the evidence provided in the literature is conflicting, mainly due to the intercorrelation between physical activity and body weight status and the methodological difficulties of establishing causal inferences. In the ISAAC Phase Three study, TV viewing for more than five hours/day was correlated with an increased risk of atopy symptoms in adolescents, and in the prospective cohort study by Byberg et al., a low physical activity level in children aged 1–4 years was associated with atopic sensitization at 13 years [49,50]. However, the reported associations were attributed to the higher body fat percentages of children with sedentary activity and the subsequent changes in adipokines levels, which are known mediators that enhance the development of atopy [51]. Our results were adjusted for the obesity status; thus, suggesting that other pathophysiological mechanisms than obesity are implicated in the complex relation between atopy and physical activity. However, when the association of the dietary patterns was taken into account, the relationship remained significant only for the current eczema symptoms status. This could imply that, for asthma and allergic rhinitis, the protective association of fruits and dairy products is of greater importance compared to the physical activity status for the down-regulation of all the inflammatory processes taking place in the development of atopic symptoms. Furthermore, vigorous physical activity has been associated with higher rates of eczema symptoms in the past 12 months in the ISAAC study, probably due to the increase in natural killer cell cytotoxicity and cytokines production, which has been observed in patients with atopic diseases [52]. However, our study assessed a complex physical activity pattern and not only vigorous physical activity; thus, the beneficial role of an active lifestyle could be mediated through the lesser body fat percentages observed in active adolescents. Further specific cohort studies should be designed in order to specifically assess the complicated relations between physical activity and atopy.

Our study had a cross-sectional design, so it was not able to infer causality in the reported associations. The recorded atopic and lifestyle characteristics were self-reported by the participating adolescents; thus, our measurements were prone to recall bias. However, the standardized questionnaire used in the GAN study and the presence of a trained researcher during the completion of the questionnaires minimized the size of bias. The existence of residual confounding could not be excluded, although all the well-known confounders were included in all the multivariable analyses. Finally, the study sample was enrolled in a large metropolitan area setting; thus, the overestimation of the prevalence of atopic diseases such as rhinitis versus the countryside should be noted.

## 5. Conclusions

Our study investigated the association of five lifestyle patterns (one physical activity pattern and four dietary ones) and their relative importance and documented that adolescents who were high consumers of fruits, vegetables and pulses had a reduced likelihood of eczema, allergic rhinitis and asthma. Moreover, a high dairy consumption was related to less asthma and allergic rhinitis symptoms. Although adherence to a physically active pattern and low consumption of unhealthy foods was firstly associated with a lesser atopy, when they were compared with the other lifestyle patterns their relative significance remained only for eczema and allergic rhinitis, respectively. Healthy dietary patterns and a physically active lifestyle should be recommended to all children and adolescents, but those with atopy may well gain an additional benefit. A fruit, vegetable and pulse intake should be the first lifestyle intervention every clinician and public health care worker evolving in the management of atopic adolescents should encourage and promote, followed by the consumption of dairy products. Finally, our findings urge the need for more longitudinal and intervention studies, in order to confirm these results and elucidate the complex associations among atopy, physical activity and diet in childhood.

## Figures and Tables

**Table 1 children-08-00932-t001:** Characteristics of the total sample according to adolescents’ atopic diseases’ history status (*n* = 1934).

Characteristics	Adolescents’ Asthma Symptoms in the Past 12 Months (Current Asthma)		Adolescents’ Allergic Rhinitis Symptoms in the past 12 Months (Current Rhinitis)		Adolescents’ Allergic Rash Symptoms in the Past 12 Months (Current Eczema)	
	**Yes**	**No**	**Yes**	**No**	**Yes**	**No**
Boys (*n*, %)	59 (49.2)	862 (47.5)	229 (49.5)	692 (47.0)	72 (41.6)	849 (48.2)
Adolescents’ age (years), mean (SD) ***						
Pet ownership (Yes, *n*, %)	81 (67.5)	1289 (71.1)	327 (70.6)	1043 (71.0)	113 (65.3)	1257 (71.4)
Having an older sibling (Yes, *n*, %)	4 (53.5)	1040 (57.4)	276 (59.6)	828 (56.3)	**85 (49.1)**	**1019 (57.9)**
Parental age (years), mean (SD)						
Parental atopic history (Yes, *n*, %)	**17 (14.2)**	**138 (7.6)**	**54 (11.7)**	**101 (6.9)**	14 (8.1)	141 (8.0)
Parental ever smoking (Yes, *n*, %)	**78 (65.0)**	**1004 (55.3)**	**282 (60.9)**	**800 (54.3)**	105 (61.0)	977 (55.4)
Parental educational level (*n*, %)						
*Primary*/*secondary* *	45 (37.5)	609 (33.5)	157 (33.9)	497 (33.7)	53 (30.8)	601 (34.1)
*Tertiary* **	75 (62.5)	1205 (66.5)	306 (66.1)	974 (66.3)	119 (69.2)	1163 (65.9)
Obesity category						
*Normal weight*	**71 (59.2)**	**1238 (68.2)**	302 (65.2)	1007 (68.5)	115 (66.5)	1194 (67.8)
*Overweight*/*Obese*	49 (40.8)	576 (31.8)	161 (34.8)	464 (31.5)	58 (33.5)	567 (32.2)
	**Yes**	**No**	**Yes**	**No**	**Yes**	**No**
High physically active adolescents **	67 (56.3)	906 (49.9)	247 (53.5)	726 (49.4)	100 (58.1)	873 (49.6)
Increased sedentary activity due to watching TV ***	39 (32.5)	467 (25.8)	**138 (29.8)**	**368 (25.0)**	52 (30.1)	454 (25.8)
Increased sedentary activity due to computer and internet activities ***	**70 (58.3)**	**829 (45.7) ***	222 (47.9)	677 (46.0)	79 (45.7)	820 (46.6)
High consumption of fruits per week (most or all days/week)	69 (57.5)	1219 (67.2)	294 (63.5)	994 (67.6)	119 (68.8)	1169 (66.4)
High consumption of cooked vegetables per week (most or all days/week)	18 (15.1)	273 (15.0)	**45 (9.7)**	**246 (16.7)**	26 (15.0)	265 (15.1)
High consumption of raw vegetables per week (most or all days/week)	39 (32.5)	732 (40.4)	190 (41.0)	581 (39.5)	72 (41.6)	699 (39.7)
High consumption of pulses per week (most or all days/week)	100 (83.3)	1625 (89.6)	**394 (85.1)**	**1331 (90.5)**	153 (88.4)	1572 (89.3)
High consumption of cereals per week (most or all days/week)	65 (52.5)	960 (53.0)	254 (54.9)	769 (52.3)	**109 (63.0)**	**914 (51.9)**
High consumption of bread per week (most or all days/week)	69 (57.5)	1079 (59.5)	287 (62.0)	861 (58.5)	**120 (69.4)**	**1028 (58.4)**
High consumption of pasta per week (most or all days/week)	**38 (31.7)**	**406 (22.4)**	**124 (26.8)**	**320 (21.8)**	44 (0.422)	400 (22.7)
High consumption of rice per week (most or all days/week)	11 (0.2)	176 (9.7)	48 (10.4)	139 (9.5)	17 (9.8)	170 (9.7)
High consumption of milk per week (most or all days/week)	85 (70.8)	1417 (78.1)	370 (79.9)	1132 (77.0)	140 (80.9)	1362 (77.3)
High consumption of dairy per week (most or all days/week)	69 (57.5)	1077 (59.5)	265 (57.4)	881 (60.0)	110 (63.6)	1036 (59.0)
Low consumption of fast-food per week (up to twice/week)	114 (95.0)	1786 (98.6)	454 (98.3)	1446 (98.4)	172 (99.4)	1728 (98.2)
Low consumption of sweets, candies and lollies per week (up to twice /week)	70 (58.3)	1117 (61.7)	**252 (54.5)**	**935 (63.6)**	95 (54.0)	1092 (62.1)
Low consumption of soft drinks per week (up to twice/week)	107 (89.2)	1701 (93.9)	430 (93.1)	1378 (93.8)	161 (93.1)	1647 (93.7)

* Bold numbers denote *p* < 0.05. ** Engaging in vigorous physical activity more than 3 times per week; *** more than 3 h per day/week.

**Table 2 children-08-00932-t002:** Differences in lifestyle factors between adolescents with and without history of atopic diseases in the past 12 months (*n* = 1934).

Lifestyle Factors	Adolescents’ Asthma Symptoms in the Past 12 Months (Current Asthma) (*n*, %)		*p*	Adolescents’ Allergic Rhinitis Symptoms in the Past 12 Months (Current Rhinitis) (*n*, %)		*p*	Adolescents’ Allergic Rash Symptoms in the Past 12 Months (Current Eczema) (*n*, %)		*p*
	**Yes**	**No**		**Yes**	**No**		**Yes**	**No**	
Adherence to an active physical activity lifestyle *	20 (15.0)	427 (23.7)	0.0235	27 (15.6)	420 (25.2)	0.015	118 (24.0)	329 (22.7)	0.556
High consumption of fruits, vegetables and pulses per week (most or all days)	7 (5.3)	274 (15.2))	0.003	14 (8.1)	274 (15.1)	0.012	57 (11.6)	224 (15.5)	0.035
High consumption of carbohydrates (bread, pasta and rice) per week (most or all days)	8 (6.0)	84 (4.7)	0.476	11 (6.4)	81 (4.6)	0.296	27 (5.5)	65 (4.5)	0.365
High consumption of dairy (milk, yogurt and cheese) per week (most or all days)	49 (36.8)	916 (50.7)	0.002	215 (43.8)	750 (51.8)	0.002	89 (51.4)	876 (49.6)	0.649
Low consumption of unhealthy foods (fast-food, sweets and soft drinks) per week (up to twice)	64 (48.1)	1046 (58.0)	0.027	243 (49.5)	867 (59.9)	0.001	87 (50.3)	1023 (58.0)	0.04

* Engaging in vigorous physical activity for >3 h/day plus watching TV and engaging in computer and internet activities for less than 3 h per day.

**Table 3 children-08-00932-t003:** Results from the multiple logistic regression analysis assessing the association between history of an atopic disease (asthma, rhinitis, eczema) in the past 12 months and adolescents’ lifestyle factors, adjusted for several confounders * (*n* = 1934).

Lifestyle factors	Adolescents’ Asthma Symptoms in the Past 12 Months (Current Asthma)		Adolescents’ Allergic Rhinitis Symptoms in the Past 12 Months (Current Rhinitis)		Adolescents’ Allergic Rash Symptoms in the Past 12 Months (Current Eczema)	
	Crude OR *** (95% CI)	Adjusted * OR (95% CI)	Crude OR (95% CI)	Adjusted OR (95% CI)	Crude OR (95% CI)	Adjusted OR (95% CI)
Adherence to an active physical activity lifestyle every week **						
No (reference level)	-	-				
*Yes*	**0.57 ****** **(0.35–0.93)**	**0.59** **(0.36–0.96)**	**0.59** **(0.39–0.916**	**0.61** **(0.40–0.94)**	**1.08** **(0.85–1.39**	**1.07** **(0.84–1.37)**
High consumption of fruits, vegetables and pulses per week (most or all days)						
No (reference level)	-					
*Yes*	**0.31** **(0.14–0.67)**	**0.27** **(0.11–0.56)**	**0.49** **(0.28–0.87)**	**0.45** **(0.25–0.82)**	**0.72** **(0.53–0.97)**	**0.73** **(0.53–0.99)**
High consumption of starchy products (cereal, bread, pasta and rice) per week (most or all days)						
No (reference level)						
*Yes*	**1.31** **(0.62–2.77)**	**1.25** **(0.59–2.66)**	**1.41** **(0.74–2.7)**	**1.36** **(0.71–2.62)**	**1.24** **(0.78–1.96)**	**1.2** **(0.76–1.91)**
High consumption of dairy (milk, yogurt and cheese) per week (most or all days)						
No (reference level)						
*Yes*	**0.57** **(0.39–0.81)**	**0.59** **(0.41–0.85)**	**1.08** **(0.79–1.47)**	**1.05** **(0.77–1.44)**	**0.72** **(0.59–0.89)**	**0.73** **(0.59–0.90)**
Low consumption of unhealthy foods (fast-food, sweets, lollies and soft drinks) per week (up to twice)						
*No (reference level)*	-	-	-	-	-	-
*Yes*	**0.67** **(0.47–0.96)**	**0.69** **(0.49–0.98)**	**0.73** **(0.54–0.99)**	**0.71** **(0.53–0.92)**	**0.67** **(0.53–0.81)**	**0.61** **(0.51–0.79)**

* Adjusted for sex, obesity status, parental atopic history, pet ownership, parental smoking, having an older sibling, cooking with fuels; ** engaging in vigorous physical activity for >3 h/day plus watching TV and engaging in computer and internet activities for less than 3 h per day (*n*, %); *** OR (95% CI): odds ratio (95% CI ); **** bold numbers denote *p* < 0.05.

**Table 4 children-08-00932-t004:** Results from the multiple logistic regression analysis assessing the association between symptoms of an atopic disease (asthma, rhinitis, eczema) in the past 12 months and each of the studied lifestyle factors, adjusted for several confounders and the remaining lifestyle factors * (*n* = 1934).

Lifestyle Factors	Adolescents’ Asthma Symptoms in the Past 12 Months (Current Asthma)	Adolescents’ Allergic Rhinitis Symptoms in the Past 12 Months (Current Rhinitis)	Adolescents’ Allergic Rash Symptoms in the Past 12 Months (Current Eczema)
	Adjusted * OR (95% CI) ***	Adjusted OR (95% CI)	Adjusted OR (95% CI)
Having an active physical activity lifestyle per week **			
No (reference level)	-	-	-
Yes	0.67 (0.40–1.10)	1.21 (0.94–1.55)	**0.59 (0.38–0.91) ******
High consumption of fruits, vegetables and pulses per week (most or all days)			
No (reference level)	-	-	-
Yes	**0.27 (0.12** **–0.60)**	**0.27 (0.50** **–0.96)**	**0.46 (0.25** **–0.83)**
High consumption of starchy products (bread, pasta and rice) per week (most or all days)			
No (reference level)	-	-	-
Yes	1.21 (0.57–2.60)	1.22 (0.76–1.95)	1.26 (0.63–2.38)
High consumption of dairy (milk, yogurt and cheese) per week (most or all days)			
No (reference level)	-	-	-
Yes	**0.61 (0.42–0.89)**	**0.73 (0.59** **–0.90)**	1.11 (0.81–1.52)
Low consumption of unhealthy foods (fast-food, sweets and soft drinks) per week (up to twice)			
No (reference level)	-	-	-
Yes	0.72 (0.51–1.04)	**0.65 (0.53–0.80)**	0.76 (0.55–1.05)

* Adjusted for sex, obesity status, parental atopic history, pet ownership, parental smoking, having an older sibling, cooking with fuels and all the remaining studied lifestyle factors; ** engaging in vigorous physical activity for >3 h/day plus watching TV and engaging in computer activities (school or videogaming) for less than 3 h per day (*n*, %); *** OR (95% CI): odds ratio (95% Confidence Interval), **** bold numbers denote *p* < 0.05.

## Data Availability

Upon reasonable request.

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
