# Peer review of "Exploring the Relation between Atopic Diseases and Lifestyle Patterns among Adolescents Living in Greece: Evidence from the Greek Global Asthma Network (GAN) Cross-Sectional Study"

_children, 2021, doi:10.3390/children8100932_

Round 1
Reviewer 1 Report
Have you looked to other clinical parameters such as lung function, specific IgE? eosinophils numbers?
Author Response
Reviewer’s 1 comments
Have you looked to other clinical parameters such as lung function, specific IgE? eosinophils numbers?
Reply to Reviewer’s 1 comments
We would like to thank the reviewer for his/her fruitful comments. GAN is a multicentre study with participations from 137 countries and 384 centres globally that aimed to assess the prevalence of several atopic diseases and to seek associations between them and several risk factors. The protocol of the study does not include any other clinical parameters as the ones indicated by the Reviewer. Thus, our research does not include them. However, since we would like to expand our research and we acknowledge that the assessment of clinical parameters such as lung function, IgE and eosinophils count will add merit to our research, according to the reviewer’s comment we would implement them in the future study.
Reviewer 2 Report
Introduction: I am not persuaded that the whole body of evidence suggesting a protective role of both healthy food and physical activity on asthma, rhinitis and eczema ( references 5-12 ) is methodologically strong enough to allow overall such a strong statement.
Every physician dealing with patients with severe asthma, rhinitis and eczema could easily state that no diet could have a substantial impact on their single patient’s well-being.
Furthermore, a reverse bias could be at stake as far as physical activity and asthma are related: poorly treated patients with ongoing lung inflammation could experience exercise induced bronchospasm and limit their activity. Obesity ( as the authors properly state in discussion), may also play a role.
Moreover all the evidence of the mechanism of action of effectiveness of diet and sport being related to anti-oxidative effect is referred only on a single reference ( 12) , based on a 2004 study in children between 6 and 7 years of age only (While the authors’ study is based on patients aged 13-14.). I would suggest to move some of the strong references from the discussion to the background section .
Having said that I would add a statement specifying that the “size effect” of this evidence remains to be defined.
Methods : if it not clear from reference 14 if the single questions about asthma, rhinitis and eczema are in the setting of a validated questionnaire or not. As it is written it looks as if they are not, please detail. In this setting a prevalence of allergic rhinitis of 25.3% seems slightly high for a southern country ( in Italy for example is 20% ) suggesting a possible bias from the fact that all patients were enrolled in a large metropolitan area, with higher prevalence versus countryside. I would comment this issue in discussion.
Results :
lines 217-219 state that : “ Moreover, high dietary intake of fruits, pulses and pasta (most or all day per week consumption) was significantly associated with lesser current asthma symptoms (all p <0.05) and ..”
Lines 222-224 state that: Conversely, high dietary consumption of cereal and bread were related to higher current eczema symptoms whereas high consumption of pasta was related to higher asthma symptoms (all p<0.05).
This is difficult to understand , how can pasta be associated with lesser asthma symptoms in the first sentence and with higher symptoms in the second sentence ?
Discussion : this part is well written, with a strong literature background and gives the idea of the complexity of the issues and variable at stake.
I would consider a statement saying that while healthy food and lifestyle should be recommended to all children those with atopy may well gain an additional benefit.
Author Response
Reviewer’s 2 comments
We would like to thank the reviewer for his/her time and effort put to review our manuscript in order to provide us with his/her fruitful comments which helped us to improve our manuscript. Here is our one-by-one reply to the reviewer’s comments:
Reply to the reviewer’s 2 comments:
Introduction: I am not persuaded that the whole body of evidence suggesting a protective role of both healthy food and physical activity on asthma, rhinitis and eczema ( references 5-12 ) is methodologically strong enough to allow overall such a strong statement.
Every physician dealing with patients with severe asthma, rhinitis and eczema could easily state that no diet could have a substantial impact on their single patient’s well-being.
Furthermore, a reverse bias could be at stake as far as physical activity and asthma are related: poorly treated patients with ongoing lung inflammation could experience exercise induced bronchospasm and limit their activity. Obesity ( as the authors properly state in discussion), may also play a role.
Moreover all the evidence of the mechanism of action of effectiveness of diet and sport being related to anti-oxidative effect is referred only on a single reference ( 12) , based on a 2004 study in children between 6 and 7 years of age only (While the authors’ study is based on patients aged 13-14.). I would suggest to move some of the strong references from the discussion to the background section .
Having said that I would add a statement specifying that the “size effect” of this evidence remains to be defined.
We would like to thank the reviewer for his/her fruitful and rightful comments, that really helped us to improve our Introduction part of the manuscript. According to his/her suggestions, we added two more references from the discussion part of the manuscript to the introduction part, reference 12, line 67, thus the new references for this part are [12-14]. Moreover, according to the reviewer’s recommendations, we added a statement specifying that “the magnitude of the effect of this evidence is yet to be defined”, lines 67-68.
Methods : if it not clear from reference 14 if the single questions about asthma, rhinitis and eczema are in the setting of a validated questionnaire or not. As it is written it looks as if they are not, please detail. In this setting a prevalence of allergic rhinitis of 25.3% seems slightly high for a southern country ( in Italy for example is 20% ) suggesting a possible bias from the fact that all patients were enrolled in a large metropolitan area, with higher prevalence versus countryside. I would comment this issue in discussion.
We would like to thank the reviewer for his/her fruitful comment. The core questions assessing the asthma, rhinitis and eczema symptoms used in the GAN study are built on those used in the International Study of Asthma and Allergy (ISAAC) study and they have been validated against bronchial hyper-responsiveness. We added two more references in order to support our statement.
- Busquets R, Anto J, Sunyer J, Sancho N, Vall O. Prevalence of asthma-related symptoms and bronchial responsiveness to exercise in children aged 13-14 yrs in Barcelona, Spain. European Respiratory Journal. 1996;9(10):2094-8.
- Hong SJ, Kim SW, Oh JW, Rah YH, Ahn YM, Kim KE, et al. The validity of the ISAAC written questionnaire and the ISAAC video questionnaire (AVQ 3.0) for predicting asthma associated with bronchial hyperreactivity in a group of 13-14 year old Korean schoolchildren. Journal of Korean Medical Science. 2003;18(1):48-52.
We would also like to thank the reviewer for his/her pointful remark about the high estimation of rhinitis in our sample. We added the possible bias from the enrollment of the adolescents from a large metropolitan area in the limitations part of the discussion, lines 409-411.
Results :
lines 217-219 state that : “ Moreover, high dietary intake of fruits, pulses and pasta (most or all day per week consumption) was significantly associated with lesser current asthma symptoms (all p <0.05) and ..”
Lines 222-224 state that: Conversely, high dietary consumption of cereal and bread were related to higher current eczema symptoms whereas high consumption of pasta was related to higher asthma symptoms (all p<0.05).
This is difficult to understand , how can pasta be associated with lesser asthma symptoms in the first sentence and with higher symptoms in the second sentence ?
We would like to thank the reviewer for his/her fruitful comment and we apologize for the misunderstanding. The sentences was referring to the consumption of pulses. We corrected the mistake in line 219
Discussion : this part is well written, with a strong literature background and gives the idea of the complexity of the issues and variable at stake.
I would consider a statement saying that while healthy food and lifestyle should be recommended to all children those with atopy may well gain an additional benefit.
We would like to thank the reviewer for his/her comments about our work. As rightfully noted, healthy dietary and physical activity lifestyle is beneficial for all children but for those with atopy may gain an additional benefit. We added the statement in the Conclusions part of the manuscript, line 420-422.
Reviewer 3 Report
Very correct methodology,
sample size quite exceptional
Interesting and relevant results
Author Response
Reviewer’s 3 comments:
Very correct methodology,
sample size quite exceptional
Interesting and relevant results
Reply to reviewer’s 3 comments:
We would like to thank the reviewer for his/her nice comments about our work. We are honoured to received such great comments encourage us to continue our research.
Round 2
Reviewer 1 Report
no commnts